# Dietary Supplement Use and Interactions with Tamoxifen and Aromatase Inhibitors in Breast Cancer Survivors Enrolled in Lifestyle Interventions

**DOI:** 10.3390/nu13113730

**Published:** 2021-10-22

**Authors:** Maura Harrigan, Courtney McGowan, Annette Hood, Leah M. Ferrucci, ThaiHien Nguyen, Brenda Cartmel, Fang-Yong Li, Melinda L. Irwin, Tara Sanft

**Affiliations:** 1Yale School of Public Health, Yale University, New Haven, CT 06510, USA; Courtney.McGowan@yale.edu (C.M.); leah.ferrucci@yale.edu (L.M.F.); Thaihien.nguyen@aya.yale.edu (T.N.); brenda.cartmel@yale.edu (B.C.); fangyong.li@yale.edu (F.-Y.L.); melinda.irwin@yale.edu (M.L.I.); 2Yale Cancer Center, New Haven, CT 06510, USA; Annette.Hood@yale.edu (A.H.); tara.sanft@yale.edu (T.S.); 3Yale School of Medicine, Yale University, New Haven, CT 06510, USA

**Keywords:** dietary supplements, interactions, tamoxifen, aromatase inhibitors, breast cancer survivors, natural medicine

## Abstract

The use of dietary supplements is common in the general population and even more prevalent among cancer survivors. The World Cancer Research Fund/American Institute for Cancer Research specifies that dietary supplements should not be used for cancer prevention. Several dietary supplements have potential pharmacokinetic and pharmacodynamic interactions that may change their clinical efficacy or potentiate adverse effects of the adjuvant endocrine therapy prescribed for breast cancer treatment. This analysis examined the prevalence of self-reported dietary supplement use and the potential interactions with tamoxifen and aromatase inhibitors (AIs) among breast cancer survivors enrolled in three randomized controlled trials of lifestyle interventions conducted between 2010 and 2017. The potential interactions with tamoxifen and AIs were identified using the Natural Medicine Database. Among 475 breast cancer survivors (2.9 (mean) or 2.5 (standard deviation) years from diagnosis), 393 (83%) reported using dietary supplements. A total of 108 different types of dietary supplements were reported and 36 potential adverse interactions with tamoxifen or AIs were identified. Among the 353 women taking tamoxifen or AIs, 38% were taking dietary supplements with a potential risk of interactions. We observed a high prevalence of dietary supplement use among breast cancer survivors and the potential for adverse interactions between the prescribed endocrine therapy and dietary supplements was common.

## 1. Introduction

Dietary supplements—defined by the Dietary Supplement Health and Education Act of 1994 [1] as herbal preparations, vitamins, and minerals—are commonly used, with 51% of U.S. adults using at least one dietary supplement [2]. Among cancer survivors, dietary supplement use is even more prevalent, with NHANES 2003–2016 data indicating a 70% use among this population [2]. Despite this high prevalence of use among cancer survivors, the joint World Cancer Research Fund (WCRF)/American Institute for Cancer Research (AICR) diet and exercise recommendations for cancer prevention state that “dietary supplements should not be used for cancer prevention” [3]. Supplement use for cancer prevention has not been shown to improve outcomes [4]. Furthermore, the use of dietary supplements is not associated with any improvement in the overall survival of cancer patients [5]. Additional cancer-specific nutrition guidelines recommend that supplements should not be used by cancer survivors for cancer prevention [6].

“Stacking” is a term used to describe a form of usage where multiple dietary supplements are consumed daily (for example, one patient may take vitamin D as a single nutrient, take a calcium plus vitamin D supplement (e.g., *Citracal^®^*), and also take a multivitamin that includes vitamin D). This makes assessing dietary supplement usage a challenge. because it can result in large combinations of nutrients from different products, especially when multivitamins and multiminerals are combined with single-nutrient dietary supplements. Assessing the nutrient exposures caused by a product can be complicated. For example, if a patient takes an herbal preparation with nine different nutrients in the ingredients, the potential for interactions would need to be evaluated for each nutrient [7].

Stacking and high nutrient exposure can lead to safety concerns. Many dietary supplements carry the potential risk of pharmacokinetic and pharmacodynamic interactions when taken with prescribed medications. For example, studies with tamoxifen taken alongside approved antidepressant medications that are CYP2D6 inhibitors demonstrate decreased levels of endoxifen, the active metabolite of tamoxifen [8]. These interactions may also be caused by dietary supplements, and until clinical trials can be performed to validate these supplements’ safety, some researchers suggest that dietary supplement use by cancer patients should not be recommended [9].

While studies have looked at the prevalence of dietary supplement use among breast cancer patients receiving treatments including tamoxifen [10,11], to the best of our knowledge no research has investigated the frequency with which potential interactions may occur between dietary supplements and other endocrine therapies (i.e., aromatase inhibitors (AI)). 

The purpose of this study was to evaluate the prevalence of dietary supplement use among breast cancer survivors enrolled in healthy lifestyle interventions and to identify any potentially harmful dietary supplement interactions between tamoxifen and AIs.

## 2. Materials and Methods

We conducted a cross-sectional analysis of the baseline prevalence of self-reported dietary supplement use among women treated for breast cancer who participated in several lifestyle intervention studies (NCT02109068, NCT02110641, NCT02681965, and NCT02056067) that were completed between 2010 and 2017 [12,13].

### 2.1. Participants and Recruitment

The eligibility criteria for the original studies were similar (Table 1). Eligible participants were breast cancer survivors diagnosed within the past 5 years with stage zero to three breast cancer, who had completed chemotherapy and/or radiation therapy at least 3 months before their enrollment. The women had to be physically able to exercise (i.e., be able to participate in a walking program), agree to be randomly assigned to a study group, and give informed consent to participate in all study activities. They also had to be reachable by telephone and able to communicate in English. Women were ineligible if they were pregnant or intending to become pregnant in the next year, had experienced a recent (during the past 6 months) stroke or myocardial infarction, or had any severe uncontrolled mental illness.

The breast cancer survivors were recruited between June 2010 and February 2017 via several approaches: (1) from five hospitals in Connecticut through the Rapid Case Ascertainment Shared Resource of the Yale Cancer Center, a field arm of the Connecticut Tumor Registry; (2) from self-referral via the study brochures in the Breast Center at the Smilow Cancer Hospital at Yale-New Haven; and (3) from the active recruitment of women attending the Yale Cancer Center Survivorship Clinic. The Connecticut Department of Public Health Human Investigation Committee (for NCT02056067 participants only) and the Yale School of Medicine Human Investigation Committee approved all of the procedures, including the written and verbal (via telephone) informed consent [13].

### 2.2. Collection of Self-Reported Prescription Medication and Dietary Supplement Usage

All participants completed a self-reported frequency-based prescription medication and dietary supplement questionnaire. Regular use was defined as taking the agent at least 3 times a week for at least one month prior to the time of enrollment in the study. For the medications or supplements not listed on the questionnaire, open text fields allowed the participants to write the names of the prescription medications and dietary supplements they were taking (see the sample medication supplement form, Appendix A).

A registered dietitian (RD) with a certified specialty in oncology nutrition (CSO) (MH and CM) reviewed and standardized the self-reported dietary supplements using the Dietary Supplement Label Database (DSLD) developed by the Office of Dietary Supplements at the National Institutes of Health [14]. The generic and brand name formulas not found on the DSLD were reviewed on the manufacturer’s website for each supplement fact label. Dietary supplements were then classified into categories: single nutrient, multivitamin, multimineral, and herbal preparations.

### 2.3. Collection of Other Lifestyle and Clinical Characteristics

Medical record review and self-report questionnaires were used to determine disease stage and endocrine therapy. The majority of height and weight measurements were taken during in-person baseline visits, though one study only collected self-reported height and weight measurements at the baseline (NCT02681965, *n* = 205).

### 2.4. Dietary Supplement Potential Interactions with Tamoxifen and AIs

The potential pharmacokinetic and pharmacodynamic interactions of all self-reported dietary supplements with tamoxifen and the AIs (anastrozole, letrozole, and exemestane) were identified using the Natural Medicines Database [15] by both a clinical pharmacy specialist (PharmD) specializing in oncology (AH) and a registered dietitian (RD) with a certified specialty in oncology nutrition (CSO) (MH and CM).

The stacking of nutrients from the use of multiple dietary supplements was enumerated, and proprietary formulas were broken down into their individual ingredients in order to more accurately report the nutrient exposures and assess potential interactions. Using the proprietary Natural Medicines Database interaction grading levels, only potential interaction levels of “moderate” (described as “a significant interaction or adverse outcome could occur”) and “major” (described as “a serious adverse outcome could occur”) grading were included. All research evidence grading levels were included: level A—a high-quality randomized control trial (RCT) or meta-analysis (a quantitative systematic review); level B—a nonrandomized clinical trial, non-quantitative systematic review, lower-quality RCT, clinical cohort study, case–control study, historical study, or epidemiological study; level C—consensus or expert opinion; and level D—anecdotal evidence, an in vitro or animal study, or theory-based evidence from pharmacology.

### 2.5. Statistical Analysis

The patient characteristics were summarized using the means and standard deviations or the frequencies and percentages, as appropriate. A descriptive analysis was performed to describe the baseline dietary supplement use patterns in terms of the frequency and prevalence of each type of dietary supplement. The prevalence was also examined excluding women taking only vitamin D, calcium, and multivitamins, as these supplements are frequently recommended or prescribed by physicians to treat bone health or address other nutrient gaps in the customary diet. The number of dietary supplements taken at the baseline was classified by number of pills (e.g., *Hot Plants™ for Her* was counted as one pill, even though it has multiple active ingredients).

Using the Natural Medicines Database, we checked the individual nutrients for interactions with any of the endocrine therapy medications prescribed in our population (i.e., tamoxifen, anastrozole, letrozole, and exemestane). Patients not taking tamoxifen or AI were excluded from the analysis for supplement–drug interaction. Cross tabling was used to present the extent of the interaction.

## 3. Results

### 3.1. Baseline Characteristics

The baseline characteristics were similar for the women enrolled across the studies. The women were 58.6 (9.0) (mean (standard deviation)) years old, non-Hispanic white (86%), college-educated (68%), 2.9 (2.5) years from diagnosis, and had a BMI of 31.8 (5.9) kg/m^2^. The women were diagnosed primarily with stage I breast cancer (47%) (Table 2).

### 3.2. Frequency of Dietary Supplement Usage

Among these 475 breast cancer survivors, 393 (83%) reported using dietary supplements at the baseline, with 51% of the women taking three or more individual dietary supplements and 23% taking five or more individual dietary supplements (range = 1–23) (Figure 1). Among all dietary supplement users, 108 (28%) reported taking either vitamin D, calcium, a multivitamin, or a combination of these supplements only, with 285 (73%) taking other supplements that are not traditionally prescribed or recommended by clinicians.

The 393 dietary supplement users reported a total of 108 different types of dietary supplement. A total of 26 (24%) were single nutrients, 31 (29%) were paired nutrients (these include multivitamins and multiminerals), and 51 (47%) were herbal preparations. A total of 53 (14%) dietary supplement users took a combination of single nutrients, paired nutrients, and herbal preparations. The top 10 dietary supplements are presented in Table 3.

### 3.3. Dietary Supplement Interactions with Tamoxifen and AIs

When the nutrient exposures of all 108 self-reported dietary products were analyzed, 36 individual dietary supplement ingredients had potential interactions with either tamoxifen or any AI, as identified in the Natural Medicines Database (Table 4). The dietary supplements are classified by individual ingredient and are listed in alphabetical order, with the interacting endocrine medication and mechanism of potential interaction identified in superscript footnotes. We did not include the direction of metabolism (inducers or inhibitors) because the literature was inconsistent regarding the reporting of this information. Grapefruit extract was the only supplement that was considered a potential cause of major interactions; the remaining 35 were all considered potential causes of moderate interactions. Vitamin D was the most prevalent supplement: 191 women taking either tamoxifen or an AI reported taking vitamin D. The majority of interactions involved herbal preparations (89% versus 11% involving vitamins). The frequency of the 36 interactions varied with the type of endocrine therapy used: tamoxifen interacted with 100%, exemestane and letrozole both interacted with 72%, and anastrozole interacted with 36%.

Of the 353 women taking tamoxifen or AIs at the baseline, 38% were taking dietary supplements with the potential to produce major or moderate interactions. The highest interaction-to-use ratio was seen with exemestane and the lowest with anastrozole (exemestane 82% (23/28); letrozole 73% (80/110); tamoxifen 35% (25/71); anastrozole 4% (6/144)).

## 4. Discussion

We found a high dietary supplement usage among the breast cancer survivors, with 83% of the women taking at least one dietary supplement. Over half (51%) reported taking three or more supplements. Of those on endocrine therapy, 38% were taking supplements that had at least moderate potential for interactions.

Our study found higher dietary supplement usage compared to reports in the literature. For instance, different populations including women without cancer (51%) and cancer survivors of various disease types (76%) had a lower prevalence of dietary supplement use [16]. It should be noted that our study included all dietary supplements, both those recommended by clinicians and those initiated by patients without clinician involvement. While not all medical professionals recommend the use of vitamin D, calcium, or a daily multivitamin, these supplements are commonly recommended by clinicians, and we were unable to determine whether the use of these supplements had a medical indication. Even accounting for these sometimes-prescribed supplements, our study found a high use of “nontraditional” dietary supplements (60%).

The higher dietary supplement rate reported in our study could be explained by the high education level of our study participants, which fits with the profile of higher dietary supplement usage described by Cowan et al. In their analysis of NHANES data for 2011–2014, the overall dietary supplement use by healthy adults in the U.S. was found to be higher among women (59%) than men (45%), while higher-income and food-secure populations were more likely to consume one or more dietary supplements compared to less affluent participants [17].

The volume of supplement use per individual was also high in our study, with over half (51%) taking three or more and 23% taking five or more dietary supplements. There were 12 individuals taking 10 or more dietary supplements. To our knowledge, our study is the first to report on the broad spectrum of dietary supplement use including the volume (i.e., the total number of pills) and type of supplement by breast cancer survivors. The heterogenous, comprehensive list generated from our data collection included 108 unique supplements. Other studies have typically focused on a list of pre-specified supplements [9], a single class of nutrients (i.e., antioxidants) [11], or on non-cancer populations only [18]. In a study of healthy adults in the U.S. (2003–2006), most individuals reported taking one supplement daily, 10% reported taking three daily, and 10% reported taking five daily [19]. Du et al. found that adult cancer survivors had a higher prevalence of use of any dietary supplement compared to non-cancer survivors; however, the individual number of supplements was not reported [16].

Lee et al. looked at the potential interactions of all medications—including dietary supplements—taken by 67 prostate and breast cancer subjects before, during, and after chemotherapy. Dietary supplements were involved in 56% of the potential 1747 total interactions with chemotherapy that were identified. While there was a reported increased utilization of dietary supplements after chemotherapy (51% during vs. 66% after), the interactions of dietary supplements with tamoxifen and AIs were not evaluated [20].

Several studies have identified the health problems associated with synthetic xenoestrogens that are found in various materials, including additives or contaminants in food [21,22,23]. These endocrine-disrupting chemicals have become a part of everyday life, interfere with the natural cycle of the hormones in the body, and are thought to give rise to many endocrine-related disorders, including endocrine-related cancers. In our analysis, 11/36 (31%) of the reported dietary supplement ingredients caused estrogenic activity that could further potentiate this estrogenic exposure. This may reduce the effectiveness of hormone therapy and therefore worsen patients’ prognoses.

Of the women in our study taking a prescribed endocrine therapy, 38% were taking supplements with potential moderate interaction with the endocrine therapy. It is notable that anastrozole produced the fewest interactions (36%), due to the fact that it is not metabolized through the CYP450 enzyme pathway. Anastrozole is metabolized through the N-dealkylation, hydroxylation and glucuronidation pathway, which is not a major pathway for drug interactions [24]. For women taking dietary supplements, clinicians may consider prescribing anastrozole, as it risks the least number of potential interactions.

Vitamin D was the most common dietary supplement reported in our population. This resonates with our own clinical experience, as many breast cancer survivors are taking this supplement for bone health or low vitamin D blood levels. The Natural Medicines Database lists vitamin D as risking potential moderate interactions with Level B evidence. However, the reference included in the database specifically studied vitamin D supplementation in relation to atorvastatin concentrations and cholesterol levels. This study was small (*n* = 16), and vitamin D was found to lower atorvastatin levels, which the authors concluded was a result of vitamin D inducing the CYP3A4 enzyme and increasing the clearance of drugs metabolized in this pathway [25]. Notably, the cholesterol levels were not adversely impacted. Given that vitamin D is commonly recommended in clinical practice, we conclude that more data on vitamin D and its potential to interact with endocrine therapy is needed.

This paper investigated the individual interactions of each dietary supplement, but it should be noted that stacking occurred frequently. While we were unable to calculate the total dose of dietary supplements per participant, clinicians should be more aware of stacking, as it can result in doses that are above the recommended daily allowances.

Cancer survivors do not readily discuss their dietary supplement usage. Du et al. reported that nearly half of 1355 cancer survivors used dietary supplements on their own without consulting health care providers [16]. In another study, Pouchieu et al. reported that only 2% of 1081 cancer survivors obtained advice on dietary supplement use from an RD [26]. In addition, fewer than one half of oncologists are initiating discussions with their patients about dietary supplement use, and many indicate that a lack of knowledge and education are barriers to such discussions [27].

One way to approach a review of dietary supplement usage is to begin with the premise of “first, do no harm”. It is important for clinicians to remember that dietary supplements are legally defined as a food—and so are covered under food regulation laws—but may act like pharmacologic agents in the body [28]. Some common principles can be employed when addressing dietary supplement use with patients: (1) Meet the patient where they are at (i.e., accept that high-volume supplement users have many reasons and strong beliefs about the value added by dietary supplements). (2) Conversation starters can include questions such as, “Could you tell me about the foods you eat, and whether you take any supplements?”. (3) The goals should be to maintain an ongoing assessment of usage and to reduce the use of dietary supplements that have potential interactions with cancer therapy.

This study has several limitations. Dietary supplement usage was self-reported and thus subject to recall bias. The dosages of the dietary supplements were not reported consistently, thus we could not take dose into account in our analyses. A selection bias may exist, as our participants were mostly from the northeast region and willing to enroll in lifestyle intervention studies focused on dietary-induced weight loss and exercise. Our study’s strengths include its large study population, its comprehensive reporting of dietary supplement use, and its evaluation of all dietary supplements by ingredient. To the best of our knowledge, this is the first study to examine dietary supplement use and their potential interactions with the adjuvant endocrine therapy for breast cancer survivors.

## 5. Conclusions

We observed an 83% rate of dietary supplement use among breast cancer survivors enrolled in our study, and the potential for adverse interactions between the prescribed endocrine therapies and dietary supplements was common. These findings underscore the need for further research into the interactions between dietary supplements and endocrine therapies for breast cancer. Oncologists should be aware of dietary supplement use, understand their potential interactions with endocrine therapy, and discuss and/or refer patients to an RD and pharmacist in the multi-disciplinary team.

## Figures and Tables

**Figure 1 nutrients-13-03730-f001:**
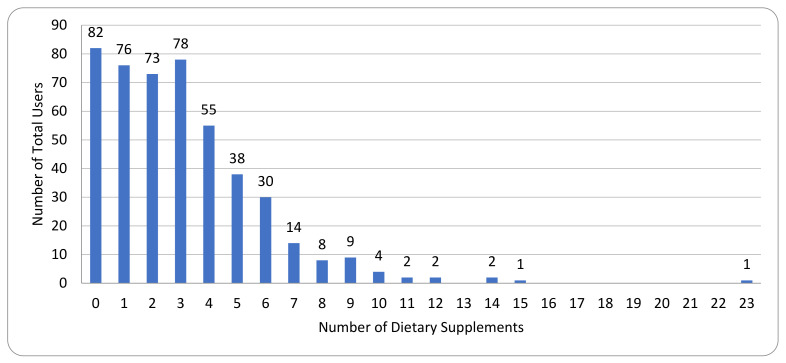
Dietary supplement use at baseline (*n* = 475).

**Table 1 nutrients-13-03730-t001:** Study eligibility criteria.

	Supervised Weight Loss Trial (NCT02109068 NCT02110641)	Self-Directed Weight Loss Trial (NCT02681965)	Supervised Exercise Trial (NCT02056067)
Study Description	6-month RCT	6-month RCT	12- month RCT
Number of Participants	151	205	119
Breast Cancer Stage	0–III	0–III	I–III
Endocrine Therapy	Tamoxifen, AI, or neither	Tamoxifen, AI, or neither	AI users only
BMI	≥25.0 kg/m^2^	≥25.0 kg/m^2^	Any BMI
Physical Activity	any amount	any amount	<90 min/week
Time Since Diagnosis	completed active treatment ≥ 3 months	completed active treatment ≥ 3 months	taking AI for 6 months to 4 years

**Table 2 nutrients-13-03730-t002:** Participant baseline characteristics.

Variable	Total Sample *n* = 475 Mean (SD) or *n* (%)
Age (years) (mean (SD))	58.6 (9.0)
BMI (kg/m^2^) (mean (SD)	31.8 (5.9)
Time since diagnosis (years)	2.9 (2.5)
Race/ethnicity	
Non-hispanic white	409 (86.1%)
Black	38 (8.0%)
Hispanic	16 (3.3%)
Other	12 (2.5)
Education	
≥College graduate	323 (68.0%)
Some school after high school	91 (19.2%)
High school graduate	57 (12.0%)
Refused to answer	4 (0.8%)
Stage	
0	52 (11.0%)
I	225 (47.4%)
II	132 (27.8%)
III	43 (9.1%)
Do not know	23 (4.8%)
Endocrine therapy usage (*n* = 475)	
None	122 (26%)
Tamoxifen	71 (15%)
Anastrozole	144 (30%)
Letrozole	110 (23%)
Exemestane	28 (6%)

SD: standard deviation.

**Table 3 nutrients-13-03730-t003:** Top 10 dietary supplements reported among the women reporting use of dietary supplements (*n* = 393).

Dietary Supplement	Participants Using Dietary Supplement *n* = 393
Vitamin D	238 (61%)
Calcium	200 (51%)
Multivitamin	198 (50%)
Omega 3	73 (19%)
Vitamin B12	68 (17%)
Vitamin C	52 (13%)
Glucosamine	42 (11%)
Fish oil	37 (9%)
Biotin	33 (9%)
Coenzyme Q10	31 (8%)

**Table 4 nutrients-13-03730-t004:** Potential Interactions with endocrine therapies.

Dietary Supplement *	Interactions with Endocrine Therapy
Astaxanthin	Tamoxifen ^1^
	Exemestane ^1^
	Letrozole ^1^
Black Cohosh	Tamoxifen ^2,3,4^
	Exemestane ^4^
	Letrozole ^4^
	Anastrozole ^4^
Boswellia serrata extract	Tamoxifen ^1,2,6^
	Exemestane ^1^
	Letrozole ^1^
Chamomile	Tamoxifen ^1,2,4,6^
	Exemestane ^1,4^
	Letrozole ^1,4^
	Anastrozole ^4^
Cinnamon	Tamoxifen ^3^
Cranberry extract	Tamoxifen ^1^
	Exemestane ^1^
	Letrozole ^1^
Diindolylmethane	Tamoxifen ^4^
	Exemestane ^4^
	Letrozole ^4^
	Anastrozole ^4^
Diosmin	Tamoxifen ^1,5,6^
	Exemestane ^1^
	Letrozole ^1^
Echinacea	Tamoxifen ^1^
	Exemestane ^1^
	Letrozole ^1^
Eleuthero	Tamoxifen ^4,5,6^
	Exemestane ^4^
	Letrozole ^4^
	Anastrozole ^4^
Garlic extract	Tamoxifen ^1^
	Exemestane ^1^
	Letrozole ^1^
Gingko biloba	Tamoxifen ^1,5,6^
	Exemestane ^1^
	Letrozole ^1^
Ginseng	Tamoxifen ^1,2,4,6,7^
	Exemestane ^1,4^
	Letrozole ^1,4^
	Anastrozole ^4^
Glucomannan	Tamoxifen ^9^
	Exemestane ^9^
	Letrozole ^9^
	Anastrozole ^9^
Grapefruit extract**	Tamoxifen ^1^**^,4,6,7^**
	Exemestane ^1^**^,4^
	Letrozole ^1^**^,4^
	Anastrozole ^4^
Grapeseed	Tamoxifen ^1,2^
	Exemestane ^1^
	Letrozole ^1^
Green tea extract	Tamoxifen ^3^
Hesperidin	Tamoxifen ^5^
Horny goat weed (Epimedium grandiflorum)	Tamoxifen ^4^
	Exemestane ^4^
	Letrozole ^4^
	Anastrozole ^4^
Jambolan (prune)	Tamoxifen ^6^
Maca root	Tamoxifen ^4^
	Exemestane ^4^
	Letrozole ^4^
	Anastrozole ^4^
Methoxylated flavones	Tamoxifen ^1,5^
	Exemestane ^1^
	Letrozole ^1^
Milk thistle	Tamoxifen ^1,4,10^
	Exemestane ^4^
	Letrozole ^4^
	Anastrozole ^4^
Niacin	Tamoxifen ^3^
Quercetin	Tamoxifen ^1,2,6^
	Exemestane ^1^
	Letrozole ^1^
Red yeast rice	Tamoxifen ^3^
Resveratrol	Tamoxifen ^1,4^
	Exemestane ^1,4^
	Letrozole ^1,4^
	Anastrozole ^4^
Rhodiola root	Tamoxifen ^5,6^
Sesame seed	Tamoxifen ^6,8^
Slippery elm bark	Tamoxifen ^9^
	Exemestane ^9^
	Letrozole ^9^
	Anastrozole ^9^
Sulforaphane	Tamoxifen ^1^
	Exemestane ^1^
	Letrozole ^1^
Sweet orange	Tamoxifen ^5^
Turmeric extract	Tamoxifen ^1,3,4^
	Exemestane ^1,4^
	Letrozole ^1,4^
	Anastrozole ^4^
Vitamin A	Tamoxifen ^3^
Vitamin D	Tamoxifen ^1^
	Exemestane ^1^
	Letrozole ^1^
Vitamin E	Tamoxifen ^1^
	Exemestane ^1^
	Letrozole ^1^

* Classified by individual ingredient; ** indicates major interaction, otherwise all interactions listed below are moderate; ^1^ CYP3A4; ^2^ CYP2D6; ^3^ pharmacodynamic: hepatotoxic; ^4^ pharmacodynamic: estrogenic activity; ^5^ P-glycoprotein substrates; ^6^ CYP2C9; ^7^ may increase the effect of the drug; ^8^ pharmacodynamic: decreases the tumor inhibitory effect of tamoxifen; ^9^ decreases drug absorption; ^10^ inhibits UGT, causing decreased drug clearance.

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
