# Peer review of "Dietary Supplement Use and Interactions with Tamoxifen and Aromatase Inhibitors in Breast Cancer Survivors Enrolled in Lifestyle Interventions"

_nutrients, 2021, doi:10.3390/nu13113730_

Round 1

Reviewer 1 Report

In the present manuscript Maura Harrigan and collaborators analyze the frequency of use and characterize the typology of nutritional supplements in a sample of women who survived breast cancer in anti-estrogenic hormonal therapies (tamoxifen and aromatase inhibitors). The main finding is that a sizeable number of patients take supplements that potentially interfere with hormone therapy.

the manuscript is pleasant to read, the methods well described, and the results are well exposed and sufficiently convincing. in the discussion the limits are well defined, above all to the choice of the sample

The study, indeed, refers to a sample of women enrolled in Trials on the role of physical exercise and lifestyle, therefore it has inherent limitations in the selection of the sample (among other things, well described by the authors the type of patient "at risk" of abuse of supplements, i.e. obese / overweight and with a high level of education).

I find the paper very interesting and clinically useful above all because it raises a problem that is often overlooked in clinical practice. The only note that I would like to emphasize is that in the discussion there is no adequate incisiveness on the potential clinical implications of the use / abuse of supplements.

In this regard, I would like to advise the authors to reinforce the message:  many of the supplements shown in the table, probably used for the potential antioxidant and immuno-modulator action (eg resveratrol, to name the most popular), have an estrogenic action especially considering the potential additive effect to environmental contaminants and diets.  this could be a problem because it would reduce the effectiveness of hormone therapy and therefore worsening the prognosis. To this end, I would like to point out some potentially useful references:

Rashid H, Alqahtani SS, Alshahrani S. Diet: A Source of Endocrine Disruptors. Endocr Metab Immune Disord Drug Targets. 2020;20(5):633-645. doi: 10.2174/1871530319666191022100141. PMID: 31642798.

Paterni I, Granchi C, Minutolo F. Risks and benefits related to alimentary exposure to xenoestrogens. Crit Rev Food Sci Nutr. 2017 Nov 2;57(16):3384-3404. doi: 10.1080/10408398.2015.1126547. PMID: 26744831; PMCID: PMC6104637.

Qasem RJ. The estrogenic activity of resveratrol: a comprehensive review of in vitro and in vivo evidence and the potential for endocrine disruption. Crit Rev Toxicol. 2020 May;50(5):439-462. doi: 10.1080/10408444.2020.1762538. PMID: 32744480.

Author Response

Dear Editors and Reviewers

The comments are excellent and well taken. Our responses are listed below the reviewer's comments.

Reviewer 1 Comments: 

I find the paper very interesting and clinically useful above all because it raises a problem that is often overlooked in clinical practice. The only note that I would like to emphasize is that in the discussion there is no adequate incisiveness on the potential clinical implications of the use / abuse of supplements.

In this regard, I would like to advise the authors to reinforce the message:  many of the supplements shown in the table, probably used for the potential antioxidant and immuno-modulator action (eg resveratrol, to name the most popular), have an estrogenic action especially considering the potential additive effect to environmental contaminants and diets.  this could be a problem because it would reduce the effectiveness of hormone therapy and therefore worsening the prognosis. To this end, I would like to point out some potentially useful references:

Rashid H, Alqahtani SS, Alshahrani S. Diet: A Source of Endocrine Disruptors. Endocr Metab Immune Disord Drug Targets. 2020;20(5):633-645. doi: 10.2174/1871530319666191022100141. PMID: 31642798.

Paterni I, Granchi C, Minutolo F. Risks and benefits related to alimentary exposure to xenoestrogens. Crit Rev Food Sci Nutr. 2017 Nov 2;57(16):3384-3404. doi: 10.1080/10408398.2015.1126547. PMID: 26744831; PMCID: PMC6104637.

Qasem RJ. The estrogenic activity of resveratrol: a comprehensive review of in vitro and in vivo evidence and the potential for endocrine disruption. Crit Rev Toxicol. 2020 May;50(5):439-462. doi: 10.1080/10408444.2020.1762538. PMID: 32744480.

Response:

Thank you for this important perspective. We agree and have added content addressing the additive estrogenic effect using these 3 references. We have also added a sentence to make a more incisive statement as to the potential clinical implications of supplement interactions with Tamoxifen and AIs. (Line 236)

Reviewer 2 Report

This review is in regards to the manuscript entitled “Dietary Supplement Use and Interactions with Tamoxifen and 2 Aromatase Inhibitors in Breast Cancer Survivors Enrolled in 3 Lifestyle Interventions” by Harrigan and colleagues. The study reports on supplement interactions among breast cancer survivors on adjuvant endocrine therapy enrolled in lifestyle clinical trials. The authors report significant dietary supplement use among this cohort (83%) and conclude that over one-third of supplement users were at risk for interactions with their endocrine therapy. The manuscript is well written and the findings important for this population. A strength is the use of the Natural Medicines Comprehensive Database interaction software. The study would be enhanced by considering a more expansive evaluation of potential medication interactions as well additional references on the topic in the discussion.

Major Comments

  1. The risk of potential supplement-medication interactions is likely higher and thus an analysis of all medicines should be considered.
  2. Consider also utilizing other medication interaction software as well such as Lexicomp or Micromedex.
  3. Table 4 would be more helpful by including the result of the interaction and severity rating.
  4. Discussion would be improved by providing more comparisons to other published studies on medication interactions among cancer patients
    1. van Leewan RW, Reichelman RP, McCune JS, Lee RT

Minor Comments

Minor Comments

  1. Table 3 – include top 10 supplements
  2. Limitations – Additional thoughts
    1. all participants were from one region.
    2. Included only supplements taken 3x/wk. Sometimes Vit D taken once a week so may have missed some.

Author Response

Dear Editors and Reviewers,

The comments are excellent and well taken. Our responses are listed below each of the reviewer's comments.

Reviewer 2 Comments: 

This review is in regard to the manuscript entitled “Dietary Supplement Use and Interactions with Tamoxifen and Aromatase Inhibitors in Breast Cancer Survivors Enrolled in Lifestyle Interventions” by Harrigan and colleagues. The study reports on supplement interactions among breast cancer survivors on adjuvant endocrine therapy enrolled in lifestyle clinical trials. The authors report significant dietary supplement use among this cohort (83%) and conclude that over one-third of supplement users were at risk for interactions with their endocrine therapy. The manuscript is well written and the findings important for this population. A strength is the use of the Natural Medicines Comprehensive Database interaction software. The study would be enhanced by considering a more expansive evaluation of potential medication interactions as well additional references on the topic in the discussion.

Major Comments

Comment 1: The risk of potential supplement-medication interactions is likely higher and thus an analysis of all medicines should be considered.

Response:

Thank you for this thoughtful comment. While we agree that the medication/supplement interactions are likely greater based on additional medications that could interact with supplements (i.e. phenytoin, clarithromycin, ketoconazole), this paper focuses on the specific medications used to treat cancer and the interactions with reported supplement use. Repeating this process with the complete medication list of each participant is beyond the scope of this paper and we feel would also take away the emphasis on the cancer treatment and outcomes implications of the paper.

This study chose to focus on those medications (tamoxifen and AIs) because they are directly intended to reduce recurrence. We decided to check interactions on supplements and not other prescribed medications because 1) supplement use is underreported, making our study unique in that we rigorously collected supplement use data and 2) medication-medication interactions are often flagged in the prescription writing process (i.e. if you prescribe tamoxifen for a patient on Prozac, for instance, you will get an alert warning you of the interaction). Our study, therefore, focuses on cancer therapy and interactions with supplements, almost entirely unprescribed.  

Comment 2: Consider also utilizing other medication interaction software as well such as Lexicomp or Micromedex.

Response: Thank you so much for this suggestion, we did search these databases and the data retrieved were not as well-resourced for the supplements as the Natural Medicines Comprehensive Database. If additional information were found, we would have been happy to include them here. However, these databases are largely meant for drug-drug interactions which involved two prescribed medications and not drug-supplement interactions. We did have our oncology pharmacist (AH) re-do the process in these databases but the information retrieved was scarce. Therefore, we will stick with the Natural Medicines Comprehensive Database which culls the literature and includes a wide library of references, making the knowledge gained from this process with this database more complete. We could submit additional examples if you felt you wanted to review our findings but we did not modify the manuscript to reflect this search as it did not add to the content of the manuscript.

Comment 3: Table 4 would be more helpful by including the result of the interaction and severity rating.

Response: Thank you. The type of interaction and the severity rating are found in the footnotes of the table and coded by the superscript after name of the medication or starred (all interactions are moderate except those ** which are major, we did not include minor in this analysis). We had tried to make this simple and not repetitive. We did not include direction of metabolism (inducers or inhibitors) because the literature was inconsistent with reporting of this information. We felt it was difficult to draw conclusions on this. We will add a sentence explaining this in our results section (line 174).

Comment 4: Discussion would be improved by providing more comparisons to other published studies on medication interactions among cancer patients

    1. van Leewan RW, Reichelman RP, McCune JS, Lee RT

Response: We agree and have added content using this reference (line 230).

Minor Comments

  1. Table 3 – include top 10 supplements

Response: Done (line 165).

  1. Limitations – Additional thoughts
    1. all participants were from one region.

Response: Most participants were from the northeast region, added to Limitations (line 291).

    1. Included only supplements taken 3x/wk. Sometimes Vit D taken once a week so may have missed some.

Response: While this is true, the 3x/wk was how our questionnaire was originally structured. We define this in the methods and hope this is clear to the reader (line 94 and Appendix A).

In closing, we hope you look favorably on our responses. We will certainly entertain all the other comments and appreciate this thorough review which has strengthened our submission.

Thank you again for your time, interest and support of our submission.

Sincerely,

Maura Harrigan, MS, RD, CSO

Tara Sanft, MD